# Deep Learning Approach at the Edge to Detect Iron Ore Type

**DOI:** 10.3390/s22010169

**Published:** 2021-12-28

**Authors:** Emerson Klippel, Andrea Gomes Campos Bianchi, Saul Delabrida, Mateus Coelho Silva, Charles Tim Batista Garrocho, Vinicius da Silva Moreira, Ricardo Augusto Rabelo Oliveira

**Affiliations:** 1Graduate Program in Instrumentation, Control and Automation of Mining Processes, Instituto Tecnológico Vale, Federal University of Ouro Preto, Ouro Preto 35400-000, Brazil; 2VALE S.A., Parauapebas, Para 68516-000, Brazil; vinicius.silva.moreira@vale.com; 3Computing Department, Federal University of Ouro Preto, Ouro Preto 35400-000, Brazil; andrea@ufop.edu.br (A.G.C.B.); saul.delabrida@ufop.edu.br (S.D.); mateuscoelho.ccom@gmail.com (M.C.S.); ctgarrocho@gmail.com (C.T.B.G.); rrabelo@gmail.com (R.A.R.O.)

**Keywords:** edge AI, DNN, iron ore quality, AIoT

## Abstract

There is a constant risk of iron ore collapsing during its transfer between processing stages in beneficiation plants. Existing instrumentation is not only expensive but also complex and challenging to maintain. In this research, we propose using edge artificial intelligence for early detection of landslide risk based on images of iron ore transported on conveyor belts. During this work, we defined the device edge and the deep neural network model. Then, we built a prototype will to collect images that will be used for training the model. This model will be compressed for use in the device edge. This same prototype will be used for field tests of the model under operational conditions. In building the prototype, a real-time clock was used to ensure the synchronization of image records with the plant’s process information, ensuring the correct classification of images by the process specialist. The results obtained in the field tests of the prototype with an accuracy of 91% and a recall of 96% indicate the feasibility of using deep learning at the edge to detect the type of iron ore and prevent its risk of avalanche.

## 1. Introduction

In order to remain competitive in the market, the mining industries need to look for mechanisms that improve the safety of the people who work there [1]. Among other measures, this safety improvement involves the implementation of physical barriers, reduction of exposure time, improvement in operating techniques, training of operating teams, and developing sensors to detect risk conditions. A large amount of novel technologies assess this issue, such as wearable computing [2], virtual reality (VR)-based training [3], the Internet of Things (IoT) [4], and edge computing [5].

In this context, one of the operational risks in iron ore processing plants is the risk of material avalanche [6] due to changes in the physical and chemical characteristics of the iron ore extracted from the mine and sent for processing at the plant. For the ore processing, it is temporarily stored in piles or silos. These changes in the ore characteristics lead the material to develop different behaviors. That is, the material that was easily stacked or stored starts to run off violently and unexpectedly.

This issue causes tons of iron ore to travel for significant distances, destroying structures in front of them, causing material damage, and potentially reaching plant operators, causing injuries and even death. Figure 1 shows an avalanche event in an iron ore plant. The operational teams can visually recognize and classify which type of iron ore has the most significant risk of collapsing when the material is stacked or stored by conveyor belts, and here, there is an opportunity to implement a detection system for these conditions.

The central proposal of our work is the development of this detection of the risk of collapse of iron ore based on its image. Red carried iron ore with risk of avalanche has a brighter image, while material without risk tends to be more opaque; these are the main characteristics that allow the differentiation between the types of material and its identification. These differences are shown in Figure 2.

We propose using artificial intelligence (AI) to classify the risk of material collapsing from images captured in real time. The capture of images and execution of the model will be performed in Device Edge, aligned with the edge AI paradigm. From this local detection of material at risk of avalanche, alarms will be triggered through the plants’ control system, and the operational teams will carry out the necessary corrective actions to eliminate the risk conditions.

The use of technologies, such as edge AI, that allow the prediction of possible avalanches is very welcome and has immediate acceptance in the world of the mineral industry, allowing for quick decision-making and handling corrective actions following the risk predictions that these systems can provide.

It is interesting to note that edge AI technology, despite all its potential, is still in the stage of inflated expectation in Gartner’s Hype Cycle-2020 curve [7], with several academic studies on the subject, but still few developments and applications in an industrial environment, mainly in the scope of iron ore plants.

This lack of development of systems based on edge AI technology, focusing on solving problems in iron ore processing plants, opens up a whole spectrum of opportunities for applied research on this topic, especially in solutions aimed at people’s safety.

In general, the work will be developed with studies on possible deep neural network (DNN) models that can be applied in the process of identifying the material from images collected by video cameras in real time, then we will study platforms available on the market, looking for the best cost–benefit ratio, robustness, and scalability. The penultimate stage of the work will be the research or development of training for the DNN model and its conversion for use with the selected one. Finally, a prototype will be developed to carry out field tests, allowing the verification of the system’s efficiency under actual conditions.

## 2. Background

In this section, we assess the theoretical background for this work. First, we first introduce the convolutional neural networks (CNNs). Then, we discuss the concept of edge AI.

### 2.1. Convolutional Neural Networks—CNN

CNNs are commonly applied in the implementation of problem-solving involving static images or those captured in real time. Their functioning is based on how the visual cortex of animals works, with cells responding to stimuli that depend on the stimuli’s position in the visual field and cells responding to the stimuli shape (traces, dots, curves, and others) independently of the stimuli position. The first group stimulates this second category of cells in a hierarchical structure. Hubel and Wiesel discovered this behavior of the visual cortex in 1965 in experiments carried out by measuring the brain signals of sedated cats submitted to simple imaging patterns [8].

One of the first CNNs was developed in 1980 by Fukushima, with significant advances promoted by LeCun, Bottou, Bengio, and Haffner in 1998. Based on this work, still in 1998, LeCun and collaborators proposed the structure of the LeNet5 network for digits recognition in images of handwritten characters [9]. The structure of LeNet-5 is shown in Figure 3, with all its layers [10].

Inspired by the structure of LeNet5, the AlexNet model was developed by Alex Krizhevsky, Ilya Sutskever, and Geoffrey Hinton. This model had eight layers, 65,000 neurons, and 60 million trainable parameters and was the winner of the 2012 ImageNet Large-Scale Visual Recognition Challenge (ILSVRC) with an error rate of less than 16%, completely surpassing state-of-the-art results in image recognition based on computer vision. This model showed the full potential of CNNs, starting the modern era of deep learning [11].

CNN can be decomposed, for study purposes, into three parts: the input, convolutional core, and output classifier. The input image pixels go through a sequence of convolution operations (filters), activation functions, and dimensional reduction operations—polling to extract high-level features of the image. From these extracted features, the output is composed of a network feedforward that performs the classification, indicating the probability of the occurrence of the image [12].

These filters are composed of fxf pixels dimension matrices that will traverse the entire image for a number of steps depending on the wxh image dimensions, performing convolution operations between the image pixels and the filter values. The result of this process is the feature map extracted by the filter, represented by an output matrix, in Figure 4. The filter slip distance will be given by the hyperparameter *s*—stride [8]. To guarantee an integer number of filter steps, it may be necessary to fill the image edges with extra pixels, normally with values equal to zero; this hyperparameter is given by *p*—padding. The relationships between the steps executed by the filter horizontally and vertically are given by Equations (Equation 2) and (6) [9].
(1)$$horizontalrelation=\frac{\left(w-f+2p\right)}{\left(s+1\right)}$$
(2)$$verticalrelation=\frac{\left(h-f+2p\right)}{\left(s+1\right)}$$

The number of weights between the convolutional layer input and its output depends on the number of filters—*n* and the filter dimension—*f* being given by Relation (Equation 3).
(3)weightnumber=(f.f.n)

To obtain better results in the training process, convolutional filters have rectified linear units (ReLU) activation functions, so the outputs associated with the feature maps will be nonlinear [12]. The activation function is defined as indicated in Equation (Equation 4).
(4)ϕ(x)=max(0,x)

Then, the image feature maps are submitted to the subsampling process, called polling. In this process, there is a reduction in the resolution of the feature map with the addition of the semantic information contained in its pixels and the reduction of the associated spatial relevance, in other words, the grouping of pixels perceived by the feature map does not depend on its position [12]. The subsampling is carried out by applying a structure similar to the one used by filters, with the difference that neither weights nor the filling strategy on the edges of the original image are used. In this process, once the structure of the fxf subsampler is defined, f being the sampler dimension, and the distance of its displacement at each step, stride (s), it is slid over the image and the pixel set is compressed according to previously defined rules. The rules can be pixel with highest sampled value (max-pooling), pixel with lowest sampled value (min-pooling), or average of the sampled pixels (avg-pooling). The dimensions of the images compressed by the polling process are given by Equations (Equation 5) and (6), where *w*${}_{1}$ and *h*${}_{1}$ are the width and height of the feature map, respectively, and *w*${}_{2}$ and *h*${}_{2}$ are the map dimensions after compression [9].
(5)$$w_{2}=\frac{w_{1}-f}{s+1}$$
(6)$$h_{2}=\frac{h_{1}-f}{s+1}$$

### 2.2. Edge AI

The edge computing paradigm emerged from the demand for data processing as close as possible to its source, addressing problems of speed, latency, reliability, and costs of transmitting this information for centralized processing. Depending on the type of services provided by edge, they are classified as near edge, far edge, enterprise edge, and device edge [13].

Distributed processing with massive amounts of data being produced close to the physical world opens up a whole set of new opportunities for implementing AI systems. Hence, the convergence between device edge and AI models is natural and creates the AI on edge paradigm [14]. In this concept, all the power of the DNN models is used mainly in making quick inferences, without wasting time due to communication latency between the device close to the physical world, such as an instrument in the field, and the processing of the model in a central computer located in the cloud or an enterprise data center.

In implementing the AI on edge paradigm, it is necessary to consider the limitations of processing speed, energy availability, and memory size of edge devices. A roadmap for implementing AI on edge is proposed by Deng et al. [14] and considers the adaptation of existing conventional models, creation of specific training and inference frameworks, and mechanisms for accelerating the execution of the models at the hardware level.

Regarding training frameworks, these are categorized as centralized with training running on a central computer or in the cloud, decentralized with all adapted CNN training being held locally at the device edge, and hybridized with the training of a centralized model from the parameters obtained in training carried out locally; the final model is propagated to all devices [15].

For model inference, the main frameworks are edge-based mode, with the model running entirely in device edge, and edge-cloud mode with part of the model running locally and part in the cloud. For the acceleration of the DNNs models, strategies are used, such as constructing specific instruction sets for their manipulation, using parallel processors executing the models, and local memory close to the processing core.

A generic neural network accelerator architecture is shown in Figure 5, consisting of an array of specialized processors (PE) elements, each with a memory buffer to compensate for communication bus latencies. In PE, the inputs are multiplied by the weights of the network, the sum of these products is performed, and the activation functions are calculated. PEs allow the implementation of the CNN convolution, polling, and feedforward processes [16].

## 3. Related Work

Techniques with the use of computer vision, edge devices, and DNNs to detect failures in an industrial environment are proposed by several authors. In this topic, we cover some of these applications.

Cao et al. [17] proposed the usage of suspended robots to evaluate belt conveyor rollers’ thermal conditions. The robot contains sensors and a thermal camera to perform the required tasks. Although this work has some relevant aspects in edge computing, it focuses on the proposal of integration rather than the automatic detection of events of interest. Thus, their work does not cover the same aspects as this one.

Szrek et al. [18] also proposed a robot-based system to evaluate thermal conditions in belt conveyor rollers. They developed a prototype that relies on the combination of visible light and UV images to monitor the rollers. They used a controlled environment to test the features, obtaining interesting results. In the same way as the previous related work, although the context is the same, the authors cover different aspects of belt conveyor monitoring compared to this work.

Li et al. [19] displayed an intelligent monitoring system for belt conveyors in the coal industry context. For this matter, they proposed using a YOLO algorithm to detect some abnormalities, such as deviations, violations, and the presence of foreign material. Although they display a functioning system and discuss how they would apply their method in the running system, their paper lacks numerical and reproducible results, making it difficult to understand the feasibility and performance.

Detection of defects on the surface of conveyor belts is proposed with the use of computer vision combined with laser light to detect tears in the belt conveyor [20]. The images were captured using a CMOS flat matrix installed in the lower region of the belt. The laser image on the captured belt in real time was treated with suitable filters. In the simulations carried out, the system was able to detect tears quickly and accurately.

A study for the use of convolutional neural networks and images for the detection of dirt in mechanical structures of conveyor belts has been developed with promising results [21]. In this study, two network architectures were used at RsNet18 and VGG16. These were trained from 73 photographs of the clean and dirty belt structure. As the number of images was small, data augmentation techniques were used to increase the generalization capacity of the trained model. The accuracy results for identifying the presence of dirt or not was 81.8% for the ResNet18 architecture and 95.5% for the VGG16.

Failure-detection situations in the pavement are addressed using the frameworks You Only Look Once (YOLO) and Faster Region Convolutional Neural Network (Faster R-CNN) [22]. In this study, the models were trained in the identification of defects in the street asphalt. The image dataset was obtained from Google Street images. Each defect in the image was classified as belonging to one out of nine categories, classified manually by a specialist. The total number of classified images was 7237. The results were satisfactory for both the YOLO-v2 and Faster R-CNN networks with precision results equal to 93% and 75%, respectively, and F1 (overall accuracy) with values of 84% and 65%, respectively.

The feasibility of using edge AI with a DNN model to detect failures in an industrial environment is shown in the work of tear detection in conveyor belts [23]. In this study, the overall accuracy of 96% was obtained for field tests performed with the prototype built with device edge, using knowledge processor unit (KPU) to locally execute a DNN MobileNet model, trained to detect tears in the lower part of the conveyor belt. The tests were carried out in a real environment, using simulated tears in an operational conveyor belt but in the process of demobilization. Although the context and sensing unit are similar, this text monitors the ore conditions rather then the belt condition, making this context different from the previous evaluated results.

Identification of the iron ore particle size in the grinding process using images of the conveyed material on the belt and VGG model was developed, and 97% total overall accuracy results were obtained. In the model training process, four classes with 223 images were used. Transfer learning and data augmentation techniques were also used in the training stages.

## 4. Materials and Methods

To implement our research, we need to define which device edge will be used initially. In this definition, the main elements considered were the cost of the device, availability in the Brazilian market, and capabilities to run DNN models. Market research found three possible platforms: The Raspberry PI 3, Jetson Nvidia Nano, and SiPEED MAiX. The platform selected was the SiPEED. The board has small dimensions, low cost, and hardware resources ready to use convolutional neural networks. We can see the SiPEED platform in Figure 6.

### 4.1. MAiX BiT Architecture Detail

SiPEED boards have as their main component the Kendryte K210 chip, a system-on-chip (SOC) oriented towards computational vision and hearing. The Chinese company CANAAN manufactures the K210, and its block diagram can be seen in Figure 7. Its processing core comprises two 64-bit processors using an RISC-V architecture and operating at 400 MHZ, a KPU for DNN acceleration, an APU for audio processing, an FFT for Fourier transform, an SRAM with 8 MB, Timer, and a UART, 48 I/O GPIO pins, among other communication and interface devices.

In our development, we installed MAiX BiT with MicroPython firmware developed by SiPEED. This firmware allows regular operation of the K210 CPU in addition to enabling the operation of the KPU and allowing access to its functionality through specific methods of the library called MAiX [24]. With this firmware, it is also possible to use the MAiXPy IDE to connect to SiPEED and perform programming and online debugging of the source code developed in MicroPython.

The K210’s DNN hardware accelerator is the KPU. The KPU is a general-purpose processor for neural networks with low power consumption. This processor is designed to classify images in real time from previously loaded neural models. The DNN model needs to be previously compiled for use on the KPU. This process is carried out by the nncase compiler, developed by Canaan Inc.

According to the manufacturer’s manual [25], the main features of the KPU are as follows:Supports the fixed training model that the common training framework trains according to specific restriction rules.There is no direct limit on the number of network layers, which supports separate configuration of each layer of convolutional neural network parameters, including the number of input and output channels, input and output line width, and column height.Support for two convolution kernels 1 × 1 and 3 × 3.Support for any form of activation function.The maximum supported neural network parameter size in real-time work is 5.5 MiB to 5.9 MiB. The maximum support network parameter size when working in non-real-time is flash capacity.

### 4.2. AI Model Selection

After defining which edge AI would be used in the project, we selected the appropriate DNN model. The selection was performed using the compiler’s compatibility with the model and the model’s ability to identify small details and nuances in the classified image.

The architecture selected by us was MobileNet, based on the study carried out by Howard et al. [26], which demonstrates the model’s capacity for fine-grained recognition when compared to the benchmark at the time for this type of classification, in this case, the Inception v3 model. In tests conducted in the study, using the Stanford Dogs dataset, version 0.75, MobileNet-224 achieved an accuracy of 83.3% against 84.3% accuracy of Inception. Another aspect demonstrated was the significant difference between the number of parameters of the models and the consequent computational demand. Inception needs 23.2 million parameters, while MobileNet needs 3.3 million. This architecture is compatible with the nncase compiler and consequently with the SiPEED MAiX BiT KPU.

MobileNet’s architecture is composed of 28 layers. On MobileNet, conventional convolutional filters were replaced by two layers. One is called depthwise convolution, and another is called pointwise, with the effect of drastically reducing the size of the model and its computational demand [26].

Figure 8 shows the structure of the model. It is important to note that the output layer is composed of a classifier with a softmax activation function, indicating the percentage of certain probability of the classification performed by the model and the number of outputs equal to the number of classes to be identified. Initially, the model had 1000 outputs. In the case of our work, the model was modified for two output classes, one indicating the situation of good iron ore, and the other bad iron ore.

In the MobileNet training process, we will use the transfer learning technique, taking advantage of DNN’s previous training with the ImageNet dataset, for the convolutional layers. The training of our model will be carried out in the feedforward layers of the output.

### 4.3. DNN Model Training and Compilation for Use with Edge AI

The edge AI ecosystem is quite fragmented, and each device requires that the conventional DNN models, after training, be converted to executable formats by their hardware. Usually, each manufacturer develops its tool to convert the trained model in frameworks like Keras, Darknet, and TensorFlow to the appropriate format. In our case, SiPEED uses the nncase compiler to convert the model to the format used in the K210 KPU.

To automate the entire training, compacting, and compilation process, we use the aXeleRate tool. This system is based on the Keras–TensorFlow platform and is composed of a set of scripts written in Python, in the format of a Jupyter notebook for execution on the Google Colaboratory Platform [27]. The aXeleRate has a modular structure allowing its use with models such as MobileNet, NASNetMobile, ResNet, and Yolo, configured as classifiers, detectors, or segmenters. The entire process configuration, including the definition of the model architecture, training epochs, learning rate, training images source, data augmentation, and backend, among others, are configured through a data dictionary read by aXeleRate before the training process starts.

The process of training, compressing, and compiling the model for use with the SiPEED KPU run by aXeleRate is shown in Figure 9. aXeleRate and its training parameters are loaded and run from a Jupyter notebook running on the Google Colab platform. Numbered arrows show the main steps in the figure. In the first step, the dataset images, stored in Google Drive, are used by the Keras framework to train the model that will be delivered in .h5 format. The second step is to compress the model obtained previously using TensorFlow, delivering it in .tflite format. The third step uses the nncase compiler, developed by the K210 manufacturer, which will run the entire process of compiling the model from the .tflite format to .kmodel, of which the format is executable by the SiPEED KPU. The last step is to record the model on a micro SD memory card for execution on the MAiX BiT KPU using a set of MicroPython methods developed by SiPEED.

### 4.4. Prototype

For image collection and field tests of the DNN model, a prototype was built with SiPEED MAiX BiT. This prototype has a power supply from 90 to 240 VAC, a relay for interfacing with the plant’s control system, and a precision RTC clock based on the DS3231 module with I2C communication. The diagram of the electronics used is shown in Figure 10.

It was necessary to use a precision RTC to ensure the correct timestamp of the photos captured during the image collection phase of the iron ore conditions.

The image capture software and the KPU usage software running the trained DNN model were developed in uPython using the MaixPy IDE. Access to K210 KPU resources is accomplished by methods from the “Maix” library developed for uPython by SiPEED. Access to camera and LCD are performed by library “media” and access to I2C bus, for connection to RTC, is achieved with methods of library “machine”; both libraries were implemented by SiPEED as well [28]. It should be noted that access to the external RTC was carried out at the level of records since no specific library was found for this communication compatible with uPython.

The entire prototype assembly was encased in a plastic housing with an articulated bracket and a magnetic base with neodymium magnets to facilitate installation and field adjustments. The prototype installed in the field can be seen in Figure 11.

### 4.5. Methodology

The methodology for carrying out the research work can be divided into three stages. First, with a test of the feasibility of using device edge and DNN for the images available in the environment. Second, with a test of the DNN model to identify variations in ore quality between dry material and wet material. Third, with field tests of the prototype built with device edge and running the DNN model trained.

The prototype was installed in the field on an input conveyor belt, which receives material from the mine and is subject to constant variations in the presence of ore to assess the ability to detect ore variations. In this location, the prototype remained installed and took pictures every 5 s with a resolution of 224 × 224 pixels, compatible with the entry of the selected MobileNet model. This image capture takes place for two consecutive days.

Then, the set of photos was separated into two classes, one with the belt containing ore and the other with the belt without ore. This set of photos made up the dataset that was used in training the model in aXeleRate. For model validation, in aXeleRate, a subset of images was separated. This same subset was used in testing the compressed model by aXeleRate, running by the SiPEED MAiX BiT KPU.

The model validation results, both in aXeleRate and SiPEED, were analyzed by us, using a confusion matrix (an example is shown in Figure 12), and classic metrics for evaluating the performance of DNN models, accuracy, precision, recall, and *F*1-score [29]. Equations (Equation 7)–(Equation 10) define these metrics based on the model detection types: true positive (*TP*), true negative (*TN*), false positive (*FP*), and false negative (*FN*).
(7)$$acurracy=\frac{TP+TN}{TP+FP+TN+FN}$$
(8)$$precision=\frac{TP}{TP+FP}$$
(9)$$recall=\frac{TP}{TP+FN}$$
(10)$$F1=\frac{2*precision*recall}{precision+recall}$$

In the second stage of the development of the work, the prototype will be installed in the field for ten days for image collection. These images will be evaluated by a specialist in the iron ore beneficiation process and will be separated into two classes, one called good ore and the other bad ore. Bad ore is the one with the potential to create avalanches during the processing steps in the plant. With these images forming the dataset, the MobileNet model will be trained and validated in aXeleRate. The model, after compilation, will also be evaluated on the KPU. We will perform the assessments using the same metrics and confusion matrix presented above.

The last stage of the research is the installation of the prototype in the field to evaluate the performance of detecting bad ore entry into the plant through the monitored conveyor belt. The evaluation will be carried out by recording the images detected as being of bad ore. These images will have a timestamp allowing this information to be crossed with the records made by the plant’s operating teams. In our study, 15 days of continuous operation were considered. Even during this stage, the prototype will continue to record photos every 5 s, with their respective timestamp, for analysis.

It is important to emphasize that all field activities were carried out under the supervision of the operational teams to ensure the physical safety of the people involved in the device installation, adjustments, and information collection. Another important aspect of this supervision is to ensure the correct positioning of the test device regarding its field of vision and possible process and maintenance interference during the period of continuous operation.

## 5. Results

A conveyor belt that receives the material directly from the mine after the first crushing step was chosen for all image captures and field tests. The material on this belt varies a lot concerning its moisture, especially considering the rainy periods in the region where the processing plant is installed [30], in this case, Southeast Pará, in Brazil.

Another critical aspect of the installation is that both natural and artificial lighting are automatically controlled, keeping the lighting relatively constant. The tests were conducted between September and October 2021. The prototype installation location is indicated in Figure 13.

### 5.1. Validation of the DNN Model and Device Edge for Detection of Iron Ore on
the Conveyor Belt

For the validation of the MobileNet model and its entire training, conversion, and compilation process by aXeleRate for use in the SiPEED KPU, images were collected in the field for two days. The total number of images captured was 34,506.

From these images, 100 images without ore presence and 100 images with ore presence were selected, comprising the two classes of the training dataset. Another ten images were separated, from each situation, to comprise the validation set.

With these images, the MobileNet model was trained on aXeleRate running on the Google Colaboratory platform. The training was performed in 10 periods in three minutes, reaching an accuracy of 95% in the tests. The training process graphic is shown in Figure 14. The number of training times was limited to 10, considering the stabilization of the accuracy value at 95% and in accordance with the early stopping technique, to prevent overfitting of the model [9].

Our validation results for the aXeleRate-trained DNN model are shown in the confusion matrix in Figure 15a. The metrics obtained in the process are shown in Table 1. In these cases, results are obtained on aXeleRate running on Google Colaboratory.

For tests of the trained and compressed model for KPU, we used another SiPEED MAiX BiT available in the lab. The same validation images as above were used, with results indicated in Figure 15b and in the respective column of the table. The test configuration can be seen in Figure 16.

### 5.2. Model Training for Ore Identification

We used the prototype to collect an image of the iron ore transported by the belt for ten days at this stage of the work. As the photos were captured at a rate of one photo every five seconds, we obtained 172,000 photos. The photos were captured with the original SiPEED camera, with a resolution of 224 × 224 pixels and color coding in RGB565, with 2 bytes. Each photo had a timestamp generated from the prototype’s RTC. It should be noted that this RTC was manually synchronized with the plant’s ERP clock when the prototype was installed in the field.

The process specialist analyzed the 172,000 photos, and from the moments of recording of bad ore conditions in the ERP, these were categorized into two classes, one as a bad ore image and the other as a good ore. In this categorization, 130 images were obtained for each condition, totaling a dataset of 260 images. Validation images with 13 good ore and 13 bad ore situations were also separated.

From this dataset, the model was trained in aXeleRate, considering ten training periods, 0.5-level dropout, and data augmentation, with horizontal and vertical inversion of the images. The result of the training was 96.15% accuracy. The results of the training process are shown in the graphic in Figure 17. Again, the number of epochs was limited to 10 iterations to prevent overfitting and maintain the generalization characteristics of the model [31].

In the same way as in the first step here, we used the validation images to test the MobileNet model trained with the dataset of good and bad ore images both in its execution in aXeleRate and in the KPU of SiPEED MAiX BiT. The results obtained are shown in confusion matrices in Figure 18, and the results of the performance indicators of the model in Table 2. In this test, we used the ROC curve for the two types of model execution with additional criteria; Figure 19 shows these results. Examples of image classification by model, running in the cloud and device edge are shown in Figure 20.

### 5.3. Good and Bad Iron Ore Detection Field Tests

For the good and bad iron ore identification tests, we installed the prototype in the field, and the timed photo capture algorithm was still active, in addition to the real-time classifier, operating at a maximum rate of 6 fps. The device was installed and operating in the field for 15 consecutive days.

The number of photos captured automatically was 259,200. In these photos, the plant specialist identified, through visual analysis, eight photos that should have been classified as bad ore but were not recognized by the classifier as such. The prototype detected 206 situations as being bad ore. Of these, the specialist identified that 187 were correct and 19 were incorrect. The prototype’s performance can be seen in the confusion matrix in Figure 21 and Table 3.

Examples of correct detections of iron ore, carried out by the prototype in the field tests, can be seen in Figure 22.

## 6. Discussion

Considering the device edge and the DNN model adopted in this work, the ability to detect ore variations was proven with the results obtained from tests with the presence or absence of ore on the belt. The results of accuracy of 100%, recall of 93%, and F1 of 96% show the feasibility of using the device and model, even with an input resolution of 224 × 224 pixels.

Another essential aspect we evaluated in this first test was the maintenance of the MobileNet model’s performance for the test conditions, even after the execution of the entire compression and compilation process to adapt to the inherent processing capacity, memory, and energy constraints to edge devices [14].

The use of MobileNet together with SiPEED MAiX BiT was validated based on tests performed with image collection for ten consecutive days, manual classification of images with good ore and bad ore, model training, and conversion, with the respective validation of inference results provided by the SiPEED KPU. In these tests, the results displayed accuracy of 92%, recall of 100%, and F1 of 96%. Here, the tests were carried out in the laboratory, using images obtained in the field.

The result of 100% in the recall, for the tests with identification of the type of ore, was analyzed by us and came from the little variance in the images obtained for the ore considered as good. It is noteworthy that we used classic techniques such as dropout and data augmentation to minimize the effects of overfitting [9,32].

For tests conducted in the field for 15 days, with the prototype using the previously trained model, the results obtained with an accuracy of 91%, recall of 96%, and F1 of 93% were satisfactory, compared to detection systems with similar techniques studied during the development of the work.

In these field tests, the rate obtained for capturing images was six fps. This rate includes the processing time spent by the KPU in the image classification process. With this rate, identification of the material on the belt will be carried out at each 0.83 m segment, more than enough for the application in question, considering that the variations in the quality of the material are not sudden and depend on the input of material coming from the mine, carried out in batch by large trucks.

## 7. Conclusions

In this work, we discussed the possibility of using deep learning through an edge AI platform to classify iron ore during its transportation in conveyor belts. This technique uses a compressed DNN model running in an edge computing device to enable the deployment of an intelligent sensor directly in the working area. A specialist further validates the data, and the positioning information is granted through a real-time clock.

This solution is designed to assess a relevant problem in the mining industry context: material avalanches. These events happen when the transported material accumulates due to its chemical and physical conditions, causing clogging and material overload. These events cause significant equipment loss and endanger the workers that are eventually in the immediate area.

The technological aspect used to employ intelligent monitoring is the edge AI. This perspective uses the main advantages of edge computing to deploy intelligent models in edge devices. These models often need to be compressed to fit in low-power and low-cost devices. These restraints are especially relevant in edge intelligent sensing devices.

Here, we presented the training and deployment of a classification model towards an edge computing device. This device was chosen due to its capability of accelerating compressed DNN models using its hardware features. We chose a classification model compatible with the proposed training pipeline. The presented model was a MobileNet, and its layers are described in Figure 8.

Our evaluation displayed promising results in the training process, later confirmed with further data. The system classifies whether the ore is present or absent, and in a second stage, classifies the ore according to its quality. The results with automatically classified images against a specialist displayed precision of 0.91, recall of 0.96, and F1-score of 0.93. These tests were deployed in the real industrial environment, indicating the immediate feasibility of this system. Future perspectives involve the construction of more prototypes and their integration in a collaborative environment within the plant.

## Figures and Tables

**Figure 1 sensors-22-00169-f001:**
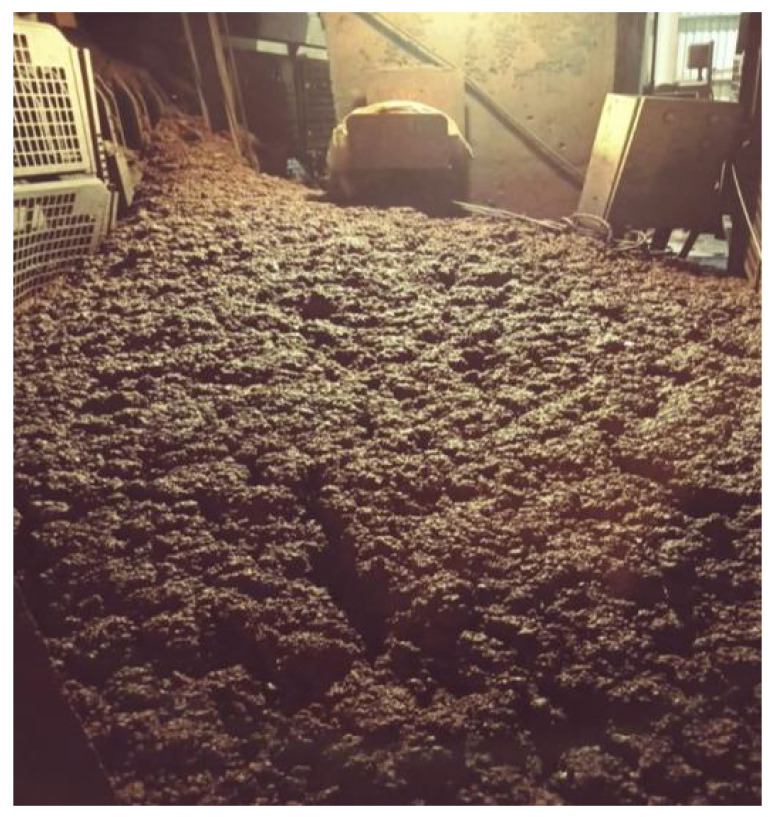
Iron ore avalanche in a beneficiation plant.

**Figure 2 sensors-22-00169-f002:**
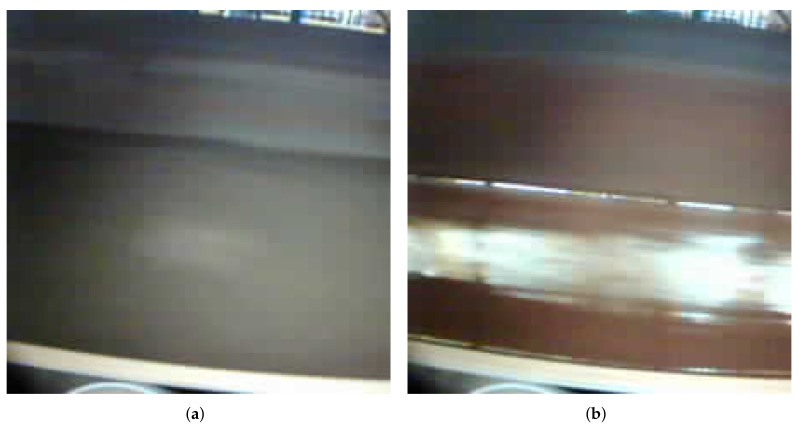
Differences in good and bad material image characteristics. (**a**) Good ore. (**b**) Bad ore.

**Figure 3 sensors-22-00169-f003:**
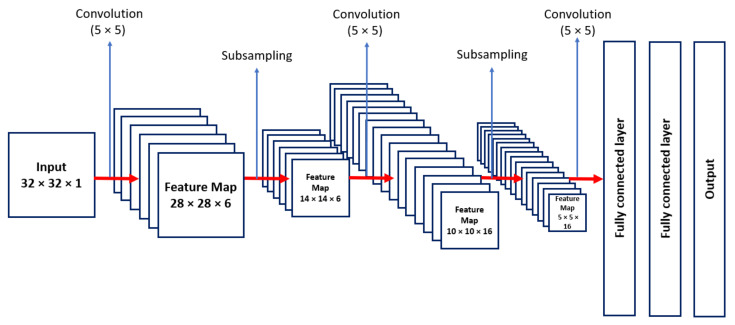
LeNet-5 convolutional neural network.

**Figure 4 sensors-22-00169-f004:**
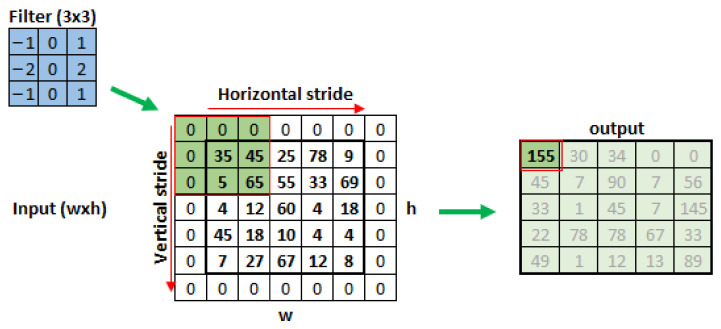
Process of extracting features from an image by convolutional filtering.

**Figure 5 sensors-22-00169-f005:**
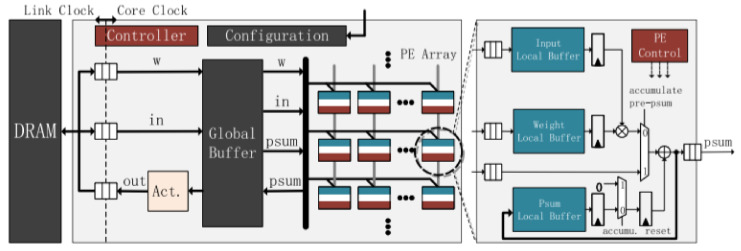
Typical architecture of accelerators for neural networks. Source: [16].

**Figure 6 sensors-22-00169-f006:**
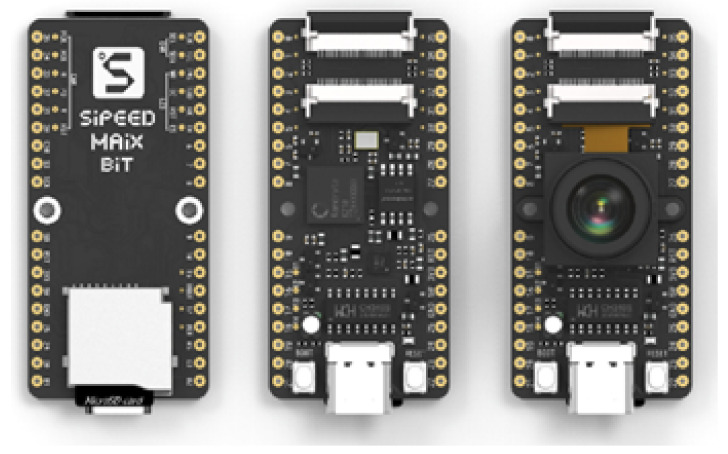
SiPEED MAiX BiT platform.

**Figure 7 sensors-22-00169-f007:**
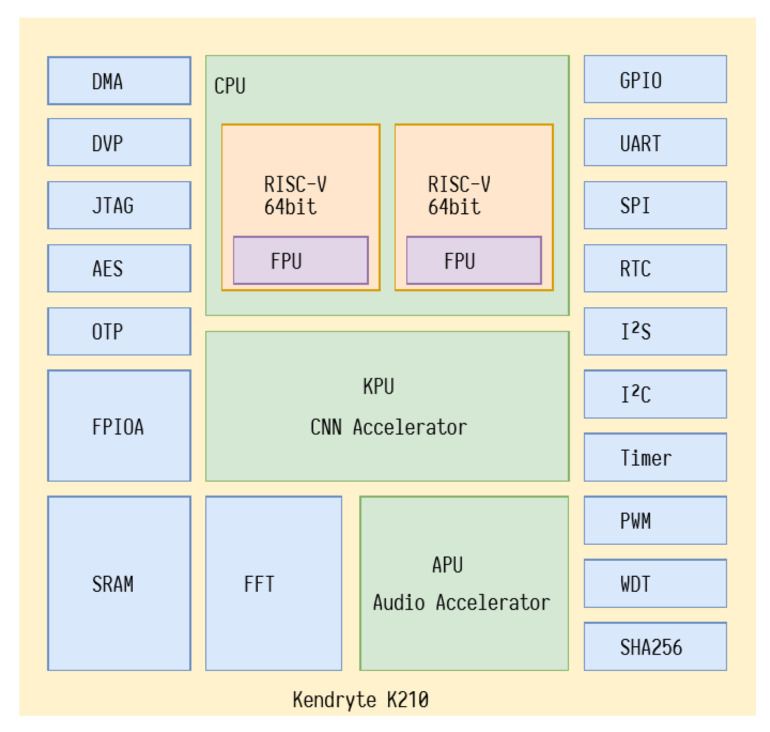
Block diagram of K210.

**Figure 8 sensors-22-00169-f008:**
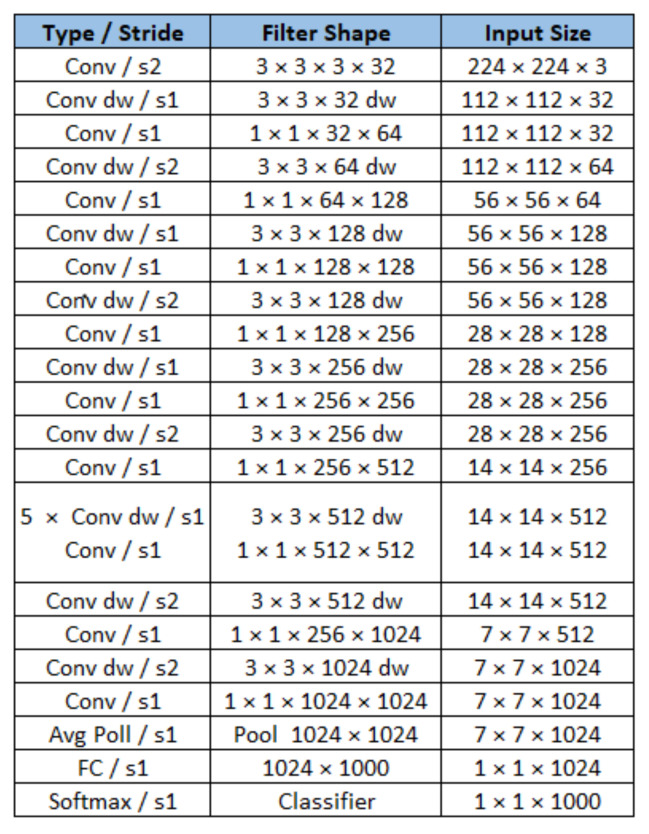
MobileNet architecture.

**Figure 9 sensors-22-00169-f009:**
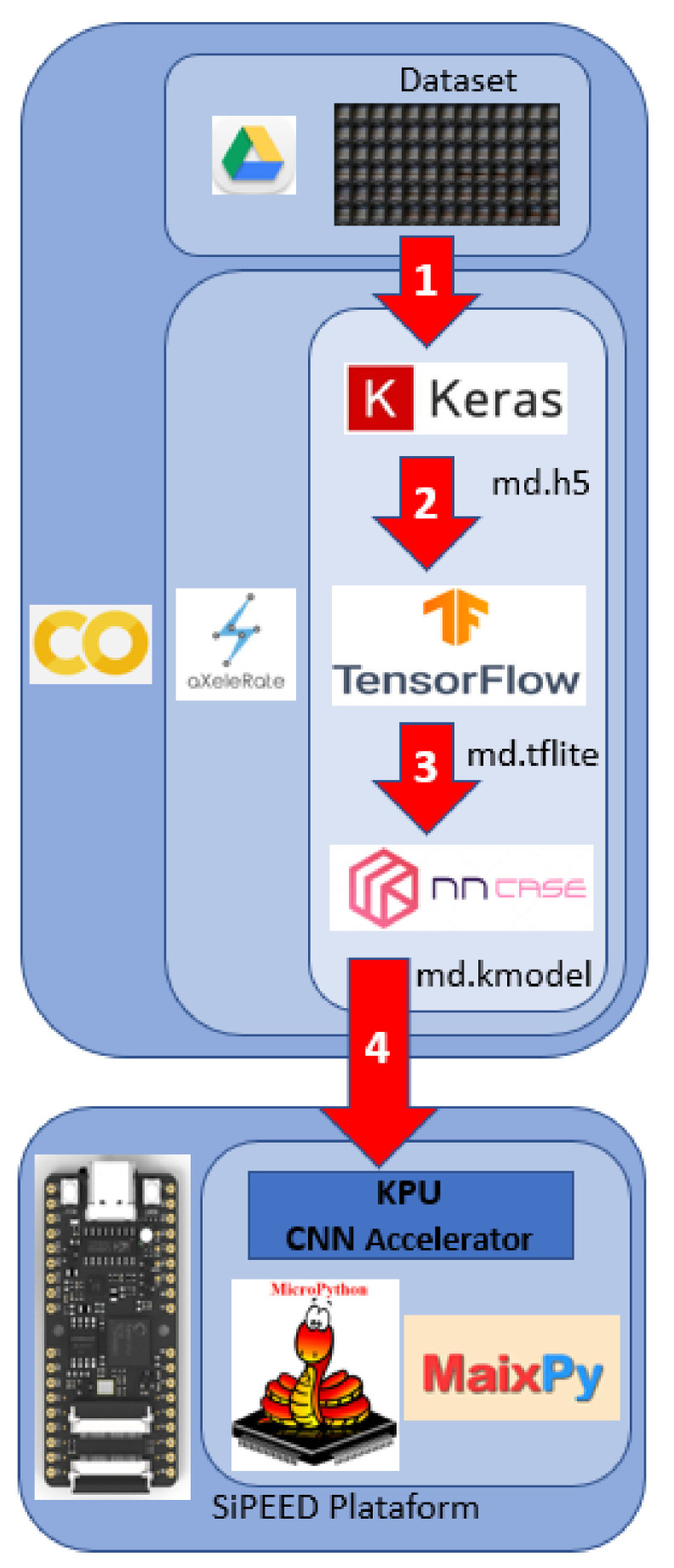
Training process with aXeleRate framework.

**Figure 10 sensors-22-00169-f010:**
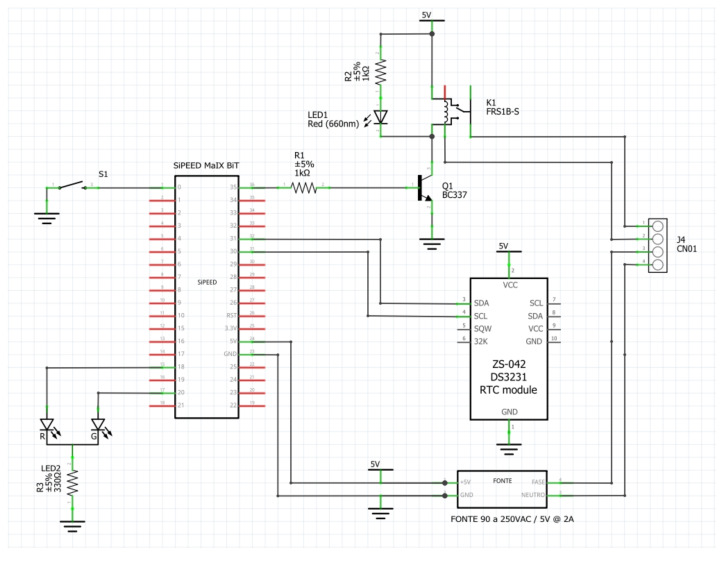
Prototype electronic diagram.

**Figure 11 sensors-22-00169-f011:**
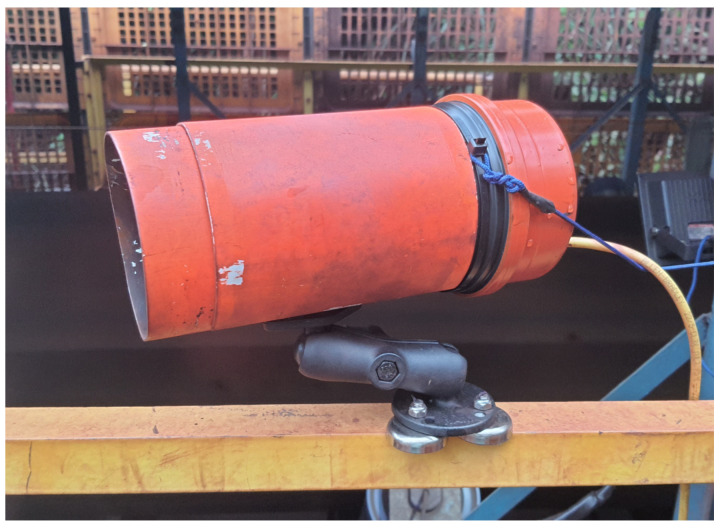
Prototype with protection and magnetic base.

**Figure 12 sensors-22-00169-f012:**
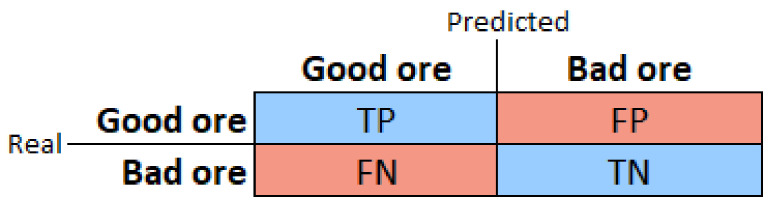
Confusion matrix.

**Figure 13 sensors-22-00169-f013:**
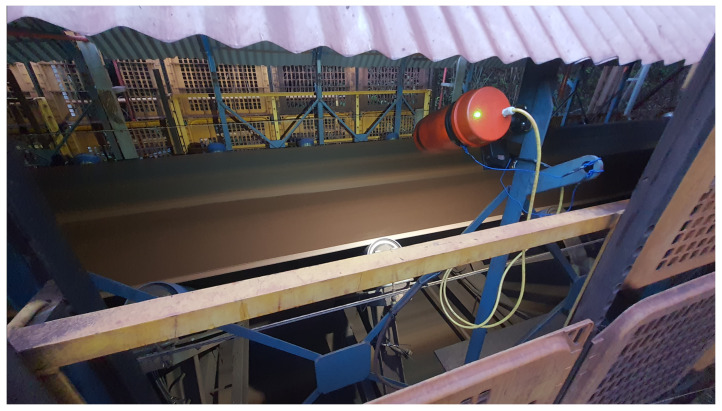
Prototype installation in the field.

**Figure 14 sensors-22-00169-f014:**
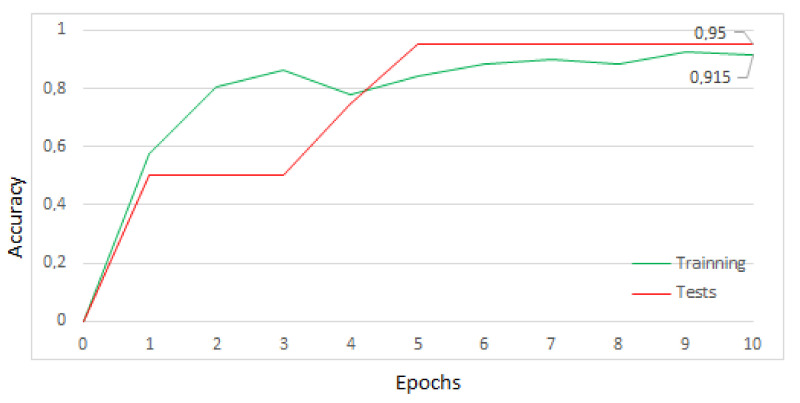
Result training with/without ore.

**Figure 15 sensors-22-00169-f015:**
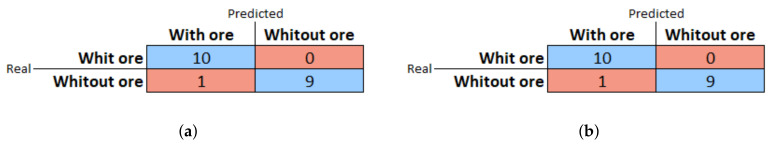
Confusion matrix for detecting the presence of ore.(**a**) aXeleRate. (**b**) SiPEED BiT.

**Figure 16 sensors-22-00169-f016:**
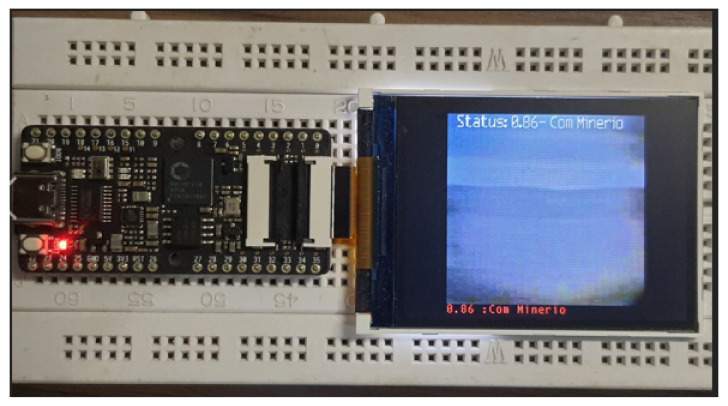
Platform to test DNN running in KPU (with ore).

**Figure 17 sensors-22-00169-f017:**
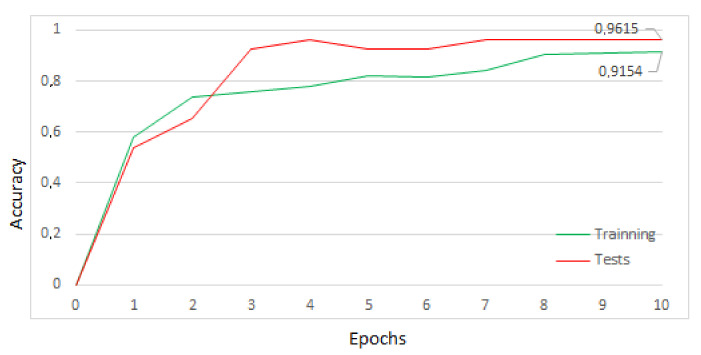
Results of training for good or bad iron ore.

**Figure 18 sensors-22-00169-f018:**
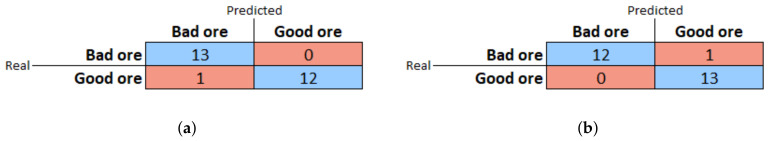
Confusion matrix for detecting the good or bad ore. (**a**) aXeleRate. (**b**) SiPEED BiT.

**Figure 19 sensors-22-00169-f019:**
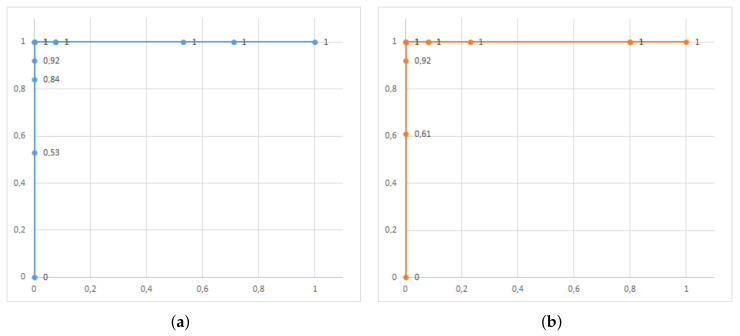
ROC curve for trained model validation images. (**a**) aXeleRate. (**b**) SiPEED BiT.

**Figure 20 sensors-22-00169-f020:**
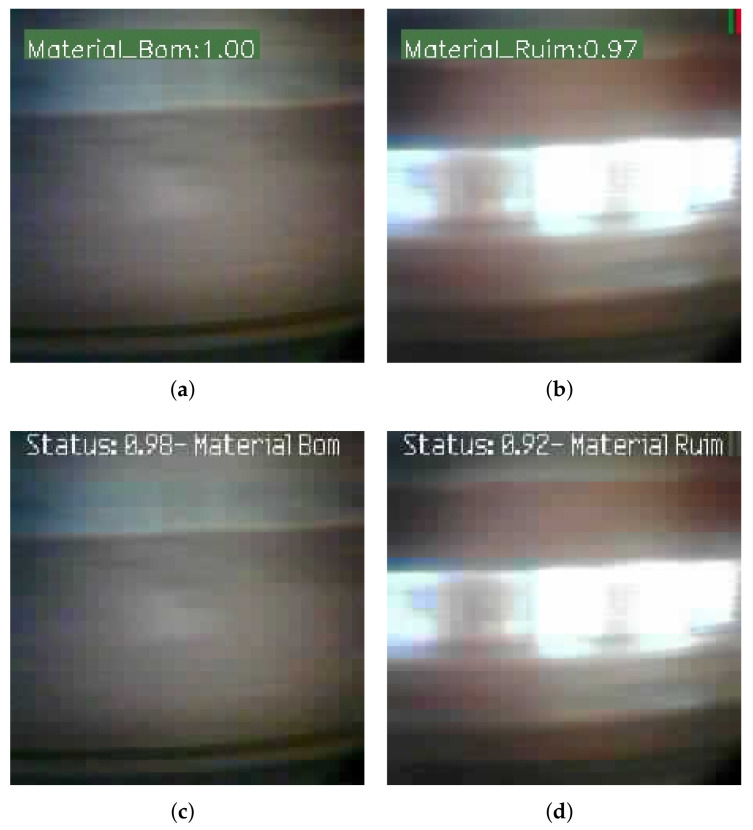
Ore images classified in aXeleRate and SiPEED. (**a**) Good Ore—aXeleRate. (**b**) Bad Ore—aXelerate. (**c**) Good Ore—SiPEED. (**d**) Bad Ore—SiPEED.

**Figure 21 sensors-22-00169-f021:**
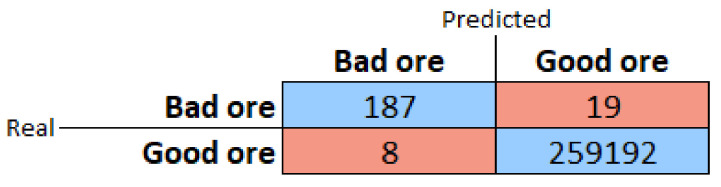
Confusion matrix for field test.

**Figure 22 sensors-22-00169-f022:**
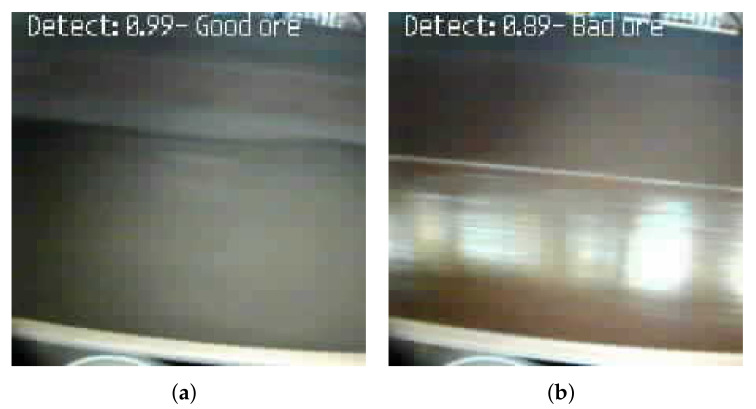
Examples of classifications performed by the prototype. (**a**) Good ore detected. (**b**) Bad ore detected.

**Table 1 sensors-22-00169-t001:** Classification results between good and bad iron ore.

Metric	aXeleRate	SiPEED BiT
Precision	1.0	1.0
Recall	0.93	0.93
F1	0.96	0.96

**Table 2 sensors-22-00169-t002:** Results obtained with the validation images, good ore and bad ore.

Metric	aXeleRate	SiPEED BiT
Precision	1.0	0.92
Recall	0.93	1.0
F1	0.96	0.96

**Table 3 sensors-22-00169-t003:** Results obtained in the detections carried out by the prototype in field tests.

Metric	Prototype—SiPEED
Precision	0.91
Recall	0.96
F1	0.93

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
