# Peer review of "Deep Learning Approach at the Edge to Detect Iron Ore Type"

_sensors, 2021, doi:10.3390/s22010169_

Round 1

Reviewer 1 Report

In this study, Klippel et al. developed the Edge artificial intelligence and the deep neural network models for early detection of landslide risk utilizing the images of iron ore transported on conveyor belts.

Although the topic is interesting and the utilized models are up-to-data, but the manuscript was not written well enough at all. Meanwhile, I think it is fair to provide the authors with a golden opportunity to improve the manuscript. My comments are as follows:

  • Try to modify the title to better present your work
  • Which features of the iron ore image have been utilized for classification?
  • The last sentences of the abstract should be devoted to presenting the key numerical findings. Your abstract presents no numerical results.
  • Although your study categorizes as a classification task, you did not use this technical term appropriately.
  • Page 2, Line 49: figure number is not entered. resolve it.
  • The quality of Figure 2 is not enough. Furthermore, you do not permit to directly utilize that. This figure may be subjected to copyright. Try to re-draw it and enhance the quality too.
  • Present the mathematical formulations of the CNN and Edge AI.
  • Generally, your work has very small numbers of mathematical formulations.
  • Remove the red underlines that appeared in texts in Figure 8.
  • The authors are requested to justify why the numbers of epochs in Figures 13 and 17 are so small?
  • Numbers of training, testing, validation, and overall databases (i.e., image) should be clearly stated.
  • The quality of Figures 19 and 21 is not enough at all.
  • The conclusion should concisely present all findings of the study. You prepared it in an awful mode. I advise you to rewrite the conclusion section.

In summary, I think this manuscript needs high levels of improvement before it can be accepted for publication. I propose the major revision for the current version of the manuscript.

Author Response

Thanks for supporting the review.

Reviewer 2 Report

This paper proposes using edge artificial intelligence for early detection of landslide risk, using images of iron ore transported on conveyor belts. Prototypes and field tests exihbit encouraging results.

The paper is well written. The following can be improved:

  1. Figure 2 is blurred.
  2. In lines 49 and 366, the reference labels (Figure ??) are not revealed.
  3. For better visulization, the tables and figures can be centered (if required by the journal). 

Author Response

Thanks for the help with the document review.

Reviewer 3 Report

Please see the file.

Author Response

Thanks for supporting the review.

Reviewer 4 Report

an average article, where some information has already been published 

Author Response

Thanks for supporting the review.

Round 2

Reviewer 1 Report

The authors have considered my previous comments. This manuscript can now be accepted for publication.

My congratulations 

Reviewer 4 Report

I recommend accepting